# Composites Based on Cotton Microfibers Impregnated with Magnetic Liquid for Magneto-Tactile Sensors

**DOI:** 10.3390/ma16083222

**Published:** 2023-04-19

**Authors:** Ioan Bica, Gabriela-Eugenia Iacobescu

**Affiliations:** 1Advanced Environmental Research Institute, West University of Timisoara, Bd. V. Parvan, Nr. 4, 300223 Timisoara, Romania; ioan.bica@e-uvt.ro; 2Department of Physics, University of Craiova, Str. A. I. Cuza, Nr. 13, 200585 Craiova, Romania

**Keywords:** magnetite, microfibers, relative dielectric permittivity, electrical conductivity, sensors

## Abstract

In this paper, we report the preparation of two new composite materials based on cotton fibers and magnetic liquid consisting of magnetite nanoparticles and light mineral oil. Using the composites and two simple textolite plates plated with copper foil assembled with self-adhesive tape, electrical devices are manufactured. By using an original experimental setup, we measured the electrical capacitance and the loss tangent in a medium-frequency electric field superimposed on a magnetic field. We found that in the presence of the magnetic field, the electrical capacity and the electrical resistance of the device change significantly with the increase of the magnetic field, then, the electrical device is suitable to be used as a magnetic sensor. Furthermore, the electrical response functions of the sensor, for fixed values of the magnetic flux density, change linearly with the increase in the value of the mechanical deformation stress, which gives it a tactile function. When applying mechanical stresses of fixed values, by increasing the value of the magnetic flux density, the capacitive and resistive functions of the electrical device change significantly. So, by using the external magnetic field, the sensitivity of the magneto-tactile sensor increases, therefore the electrical response of this device can be amplified in the case of low values of mechanical tension. This makes the new composites promising candidates for the fabrication of magneto-tactile sensors.

## 1. Introduction

The sense of touch, also known as tactile perception, is an essential ability for humans to explore the surrounding environment. Organisms can obtain information through tactile perception, such as sight and hearing. How to make robots acquire tactile perception similar to human beings is one of the hot spots in scientific research. In the bionics idea, using the human skin as a model, a large number of tactile sensors have been created. Biomimetic tactile sensors are important tools for robots to perceive the external environment, namely pressure, vibration, roughness, and temperature. Tactile sensors have played an important role in medical treatments [1,2], making artificial skin [3,4], robot tactile feedback [5], and human-machine interaction [6,7]. With the discovery of new materials and the development of microelectronics, tactile sensors based on a variety of transducing mechanisms, such as resistance [8,9,10,11], capacitance [12,13,14,15], piezoelectric [16,17,18,19], and optical [20,21,22] were produced.

In recent years, magnetic sensors are developing in the direction of high sensitivity, low power consumption, small size, etc. [23]. The performance of some types of magnetic sensors have already met the requirements of tactile sensing technology [24]. Therefore, a new type of tactile sensor called magneto-tactile sensors have recently been developed [25,26,27,28,29]. Compared with tactile sensors based on other mechanisms, magneto-tactile sensors have the advantages of high sensitivity, low hysteresis, low power consumption, and easy implementation of three-dimensional sensing and remote sensing [30].

Fabrication of composite materials based on polymeric fibers has gained increased interest in recent years since their electrical properties can be drastically changed with the addition of additives or with the variation of a magnetic flux density *B* [31,32,33]. In particular, composites based on cotton fibers impregnated with a mixture of honey, carbonyl iron (CI) and silver microparticles have shown significantly improved electrical properties [34,35]. Moreover, taking into account the therapeutic effects of each component within the mixture [36,37,38,39,40], these composites are promising candidates for the fabrication of low-cost and multi-purpose medical devices. Similarly, when barium titanate microparticles are used together with CI microparticles to impregnate cotton fabrics, the dielectric and piezoelectric properties can be controlled in a magnetic field, and respectively in a compression force field [41], which makes these composites suitable for the fabrication of sensors. The common feature in all these works is the presence of CI as a magnetizable phase.

Here, our aim is to show that by impregnating cotton fibers with a magnetic liquid (ML), one can obtain a composite material with similar or improved characteristics. To this aim, in this paper we report the preparation of two types of composites, based on MLs, in quantities of 0.138 g, and 0.276 g. From each composite, an electric device (magneto-tactile sensor) is manufactured in the form of a planar electrical capacitor, mechanically very well consolidated with a self-adhesive tape. The corresponding relative dielectric permittivity and electrical conductivity are obtained by measuring the electrical capacitance and the loss tangent of the devices when placed in an electric field with the frequency of 1 kHz superimposed on a magnetic field, and under the influence of a uniform mechanical compression force *F*. It is shown that these characteristics increase dramatically by increasing the quantity of ML and the force *F*, and they can be finely tuned by varying the magnetic flux density *B*. The distinct responses provided by the composite for *F* ≤ 8 N, and for 0 ≤ *B* (mT) ≤ 400 demonstrate that the cotton fibers, together with ML, are a low-cost material useful for the fabrication of magneto-tactile sensors, and much more stable compared to those reported in refs. [34,41].

## 2. Materials and Methods

### 2.1. Fabrication of Magnetic Composites MC

The materials used for the fabrication of the composite and the electrical device are:Cotton fabric (GB), as shown in Figure 1. At room temperature and for a relative humidity of the air of 65%, GB has the density *ρ*_GB_ ≃ 1.033 g/cm^3^. The thickness of GB is 0.4 mm. ML (EFH-1 type) is produced by Ferrotec (Santa Clara, CA, USA) [42] and bought from Magneo Smart (Romania) [43]. The ML is based on a light mineral oil and magnetite nanoparticles (Fe_3_O_4_). At room temperature, the mass density of ML is *ρ*_ML_ ≃ 1.21g/cm^3^ . The saturation magnetisation is about 35 kA/m, for magnetic field intensities *H* > 500 kA/m, as shown Figure 2a. The average diameter of the particles is *d*_Fe_3_O_4__ = 11.6 nm [44], and the volume fraction of Fe_3_O_4,_ as reported in ref. [45], is *Φ*_Fe_3_O_4__ = 6.5 vol.%. The magnetization slope of ML recorded with an experimental set-up described in Ref. [46] is shown in Figure 2a.

For MC manufacture, two pieces of material are cut from GB, in the form of a square with an edge size *L* = 30 mm. The fabrics are then introduced inside two Berzelius glasses together with a volume of 2 cm^3^ of ML. After one hour, the fabrics, impregnated with ML, are deposited on absorbent papers. In order to extract the excess ML from within the fabrics, they are pressed against the filter papers. This procedure is performed such that the remaining ML is removed, while the quantities of ML impregnated in the fabrics differ by a factor of two between the composites. At the end of this step, the quantity of ML is measured with the help of an analytic balance (AXIS 60 type). The corresponding masses and volumes of both GB and ML for the two composites are listed in Table 1.

In each composite, the volume of light mineral oil is given by $V_{\mathrm{oil}}=V_{\mathrm{ML}}-\Phi_{{\mathrm{Fe}}_{3}O_{4}}V_{\mathrm{ML}}$. Then, by using the numerical values of $V_{\mathrm{ML}}$ from Table 1 in the above relation, one obtains the volumes $V_{\mathrm{oil}}$ as listed in Table 2. The volume of microfibers from within the composites is approximated by the relation: $V_{f}=V_{\mathrm{GB}}-V_{\mathrm{ML}}$, and therefore using also the values of $V_{\mathrm{GB}}$ and $V_{\mathrm{ML}}$ from Table 1, one obtains the values of $V_{f}$ as listed in Table 2. Finally, the volume of ${\mathrm{Fe}}_{3}O_{4}$ inside the ML is obtained from $V_{{\mathrm{Fe}}_{3}O_{4}}=\Phi_{{\mathrm{Fe}}_{3}O_{4}}V_{\mathrm{ML}}$. By again using the values of $V_{\mathrm{ML}}$ from Table 1, one obtains the volumes of ${\mathrm{Fe}}_{3}O_{4}$ inside each composite, as shown in Table 2. Once all these volumes are known, the corresponding volume fractions can be easily determined. Their values are also shown in Table 2.

From the above relations (and from the numerical values in Table 2) one can see that the volumes of GB remain unchanged for both composites, and the height *h* = 0.4 mm and surface area *L*^2^ = 9 cm^2^ are also unchanged. 

The fabrics visualized at the microscope have the morphology shown in Figure 1a, while their morphology after impregnation with the magnetic liquid ML is shown in Figure 1b. The morphology of an individual thread of the fabric with ML is shown in Figure 1c. The images indicate that the cotton thread consists of many fibers of micrometric dimensions. Within the thread, the magnetic liquid ML is absorbed in the surface of the microfibers, and respectively in the interstitial space. Figure 1 further shows the existence of darker and lighter regions within the thread. This is the result of different degrees of absorption of ML, and we consider that the darker regions arise from the agglomeration of ${\mathrm{Fe}}_{3}O_{4}$ particles due to their spontaneous magnetization.

In order to highlight the surface morphologies and elemental compositions of the obtained samples, an electronic scanning microscope Inspect S (SEM) from FEI Europe B.V. (Eindhoven, Netherlands), equipped with an X-ray energy-dispersive spectrometer (EDX) was used (Figure 3). All studied samples were analyzed in low vacuum mode using the LFD detector, with a spot value of 3.5, a pressure of 30 Pa, and a high voltage of 30 kV.

The SEM microscopy shown in Figure 3a highlights the distribution of the cotton microfibers soaked with the magnetic liquid. The spectral analysis of MC composites, from Figure 3b, reveals the existence of Fe atoms together with C and O atoms from the cotton fabric and Ca atoms from the base liquid.

Between the saturation magnetization $M_{S_{\mathrm{ML}}}$ of the magnetic liquid, and the saturation magnetization $M_{S_{\mathrm{MC}}}$ of the composite, the following relation holds [47]: $\mu_{0}M_{S_{\mathrm{MC}}}=\Phi_{{\mathrm{Fe}}_{3}O_{4}}\mu_{0}M_{S_{\mathrm{ML}}}$, where $\mu_{0}$ is the vacuum magnetic constant, and the values of $\Phi_{{\mathrm{Fe}}_{3}O_{4}}$ are listed in Table 2. Based on this equation and the data in Table 2, we obtain the MC magnetization plots as shown in Figure 2b.

From Figure 2b it can be seen that the magnetization *M* of the MC increases linearly with the increase of *H* up to H=200 kA/m. Above this value of *H*, the increase of *M* is slow. For values H=500 kA/m, the magnetization tends to saturation. The magnetization *M*, for the same values of *H*, decreases with the decrease in the amount of ML used in the composites, as can be seen in Figure 2b. As such, the saturation magnetization of the composite MC_1_ is $M_{S_{\mathrm{MC}1}}=0.77\;\mathrm{kA}/m$, while for the composite MC_2_, $M_{S_{\mathrm{MC}2}}=1.54\;\mathrm{kA}/m$.

### 2.2. Fabrication of the Electrical Devices EDs

For the fabrication of the electrical devices EDs, the following materials are used:A simple textolite plate (PCu), coated with copper on one side, and with dimensions 100 × 75 × 0.8 mm^3^, from Electronic Light Tech (Bucuresti, Romania) [48]. The PCu is based on epoxy resin (FR4 type) reinforced with fiberglass. The thickness of the copper layer is 35 µm.Composites MC with dimensions 30 × 30 × 0.4 mm^3^.Patch on textile support (PTS), type Omniplast, bought from S. C. Hartmann S. R. L. (Bucuresti, Romania) [49]. The patch is a self-adhesive tape with a width of 5 cm, a thickness of 0.22 mm, and a length of 20 m.

The main steps taken for the fabrication of EDs are: From the PCu are cut two identical pieces, each one with dimensions 30 × 30 × 0.8 mm^3^.On the copper-side of each plate, two copper conductors are attached by hot-welding.The PCu plates and MC composite are arranged as follows: a film of magnetic liquid is deposited on the copper side of the PCu plates; between the PCu plates, prepared in this way, the MC composite is deposited by pressing, so that there is no air layer between the solid plate and MC. We weighed the assemblies made in this way to keep the ratio between the amounts of magnetic liquid as in Table 1.From the PTS are cut two tapes of dimensions 10 × 5 mm^2^, which are then applied to the above-mentioned system. The whole system is then consolidated by pressing and sticking the PTS tapes. A minimum 1.5 mm thick layer is formed on each side of the ED device. As such, good electrical contact is achieved between the copper foil of PCu plates and the MC. One considers that the electrical contact is established when the electrical resistance of the terminals of ED, measured with the ohmmeter UT-60, is not infinite. At the end of this step, one obtains the electrical device as shown in Figure 4a,b. It has a unitary structure resistant to medium-intensity mechanical actions (falls, hits, etc.).

### 2.3. Experimental Set-Up and Measurements

The overall configuration of the experimental setup used is shown in Figure 5.

The setup consists of an electromagnet, a continuous current source (DCS), a bridge Br, a gaussmeter Gs with a hall probe h, and a system for the application of deformation forces on the ED. The DCS current source is of type RXN-3030D produced by HAOXIN (China), the bridge Br is an RLC-meter, type CHY 41R601 (Lodz, Poland), and the gaussmeter are of type DX-102 from Dexing Magnet Tech Co. (Xiamen, China) [50]. 

The system for mechanical deformations consists of nonmagnetic components, and it has an axis that passes through the N pole of the electromagnet and which is mechanically coupled to a disk and a balanced plate. The mass on the plate consists of lead powder (pos. 6 in Figure 5). The values of the masses correspond to weight forces of 4, and respectively 8 N. The working frequency of the bridge is *f* = 1 kHz. The static magnetic field has values *B* of the magnetic flux density, of a maximum of 400 mT, tuneable in steps of 50 mT. At the beginning, and respectively during the measurements, the values of *B* are fixed between the limits ±2%. With the help of the bridge Br, one measures the electrical capacitance in series (*C*_S_) and the values of loss tangent (*D*_S_) of the device ED at about 30 s after the values *B* of the magnetic flux density are fixed. The measurements of the electrical characteristics are performed as a function of the values *B* of the magnetic flux density, superimposed on a medium-frequency electric field, and having as a parameter the force *F*, with values given by the marked masses. 

## 3. Experimental Results and Discussion

The measured values of *C*_S_ for the two composites MC_1_ and MC_2_ are presented in Figure 6a, and respectively, Figure 6b. The results show that for a fixed value of *B*, *C*_S_ slightly increases with the magnetic flux density *B*. However, for a fixed value of *B*, *C*_S_ increases more pronounced with the force *F*. In addition, the values of *C*_S_ increase with increasing the quantity of ML in the composite. 

Similar results are obtained for the loss tangent *D*_S_ (Figure 7). However, here, for a fixed value of *F*, *D*_S_ reaches first a minimum at *B* ≃ 200 mT, and then it slightly increases back to the initial values at *B* = 0 mT. The measurements of the quantities *C*_S_ and *D*_S_ when the values of B decrease do not overlap the initial ones. Deviations of up to ±4% appear between them. When repeating the measurements at 24-h intervals, the values of the measured quantities fall within the precision class of the bridge type CHY 41R 601.

The obtained results lead us to the conclusion that the equivalent electrical scheme consists of a capacitor of capacitance *C*_S_ in series with a resistor of resistance *R*_S_.

Among these quantities, the following relation holds $R_{S}={\left(2\pi fC_{S}D_{S}\right)}^{-1}$. Since the range of *C*_S_ shown in Figure 6 is of the order of pF, then the expression for the resistance can be written as:(1)$$R_{S}\left(M\Omega \right)=\frac{10^{3}}{2\pi f\left(\mathrm{kHz}\right)C_{S}\left(\mathrm{pF}\right)D_{S}}$$
where *f* is the frequency of the electric field, and *C*_S_ and *D*_S_ are the equivalent capacitance and resistance of ED, respectively.

By using the value *f* = 1 kHz, and the values *C*_S_ and *D*_S_ from Figure 6, and Figure 7, respectively, one obtains, in Figure 8a,b, the variation of resistance with magnetic flux density for the composites MC_1_, and MC_2_, respectively.

To describe quantitatively the equivalent electrical capacitance and resistance, one can start with the expressions: (2)$$C_{S}=C_{S_{0}}{\left[1-\left(2.25\Phi_{{\mathrm{Fe}}_{3}O_{4}}L^{2}B^{2}/\mu_{0}kd_{{\mathrm{Fe}}_{3}O_{4}}\right)+F/(kh_{0})\right]}^{-1}$$
and
(3)$$R_{S}=R_{S_{0}}\left[1-\left(2.25\Phi_{{\mathrm{Fe}}_{3}O_{4}}L^{2}B^{2}/\mu_{0}kd_{{\mathrm{Fe}}_{3}O_{4}}\right)+F/(kh_{0})\right]$$
which are adapted from ref. [41] by taking into account the principle of forces superposition. Here $C_{S_{0}}\equiv \varepsilon_{0}\varepsilon^{\prime }L^{2}/h_{0}$ and $R_{S_{0}}\equiv h_{0}/\left(\sigma_{0}L^{2}\right)$ are the electrical capacitance, and respectively resistance of ED at *F* = 0 N and *B* = 0 mT, $\varepsilon_{0}$ is the vacuum dielectric constant, $\varepsilon^{\prime }$ is the relative dielectric permittivity, $L^{2}$ is the common area of the electrodes-composite, $h_{0}$ is the composite’s thickness, $\sigma_{0}$ is the electrical conductivity, $\Phi_{{\mathrm{Fe}}_{3}O_{4}}$ is the volume fraction of ${\mathrm{Fe}}_{3}O_{4}$, $d_{{\mathrm{Fe}}_{3}O_{4}}$ is the average diameter of magnetite nanoparticles, and k is the coupling constant between the magnetizable phase and cotton microfibers.

Then, taking into account the model developed in ref. [51], the equivalent electrical capacitance and resistance at *F* = 0 N, and *B* = 0 mT, adapted to data obtained here, become $C_{S_{0}}=0.25\;\varepsilon_{0}\varepsilon^{\prime }L^{2}\Phi_{{\mathrm{Fe}}_{3}O_{4}}^{4/3}/d_{{\mathrm{Fe}}_{3}O_{4}}$, and respectively $R_{S_{0}}=2/\left(3\sigma_{0}L^{2}\Phi_{{\mathrm{Fe}}_{3}O_{4}}^{4/3}d_{{\mathrm{Fe}}_{3}O_{4}}\right)$. These relations show that $C_{S_{0}}$ increases while $R_{S_{0}}$ decreases with increasing the quantity of Fe_3_O_4_ particles, as confirmed by experimental data shown in Figure 6 and Figure 8.

From the expression of the electrical capacitance of the capacitor, one obtains the relative dielectric permittivity $\varepsilon^{\prime }=C_{S}h_{0}/\varepsilon_{0}L^{2}$. Then, by introducing numerical values $h_{0}=0.40\;\mathrm{mm}$, $\varepsilon_{0}=8.854\;\mathrm{pF}/m$ and $L^{2}=9{\;\mathrm{cm}}^{2}$, one obtains: (4)$$\varepsilon^{\prime }=0.0502\times C_{S}\left(\mathrm{pF}\right)$$

Thus, by using the data of *C*_S_ in Equation (4), one obtains the variation of the relative dielectric permittivity with the magnetic flux density *B*, at fixed values of force *F*, i.e., $\varepsilon^{\prime }=\varepsilon^{\prime }{\left(B\right)}_{F}$, as shown in Figure 9. 

In order to quantify the influence of the magnetic field at *F* = 0 N, on the relative dielectric permittivity, for each quantity of Fe_3_O_4_, we introduce the relative apparent magnetodielectric effect (*MDE*), as:(5)$$MDE\left(\%\right)={\left[\frac{\varepsilon^{\prime }\left(B\right)}{\varepsilon^{\prime }\left(0\right)}-1\right]}_{F=0\;N}\times 100$$
where $\varepsilon^{\prime }\left(B\right)$ is the relative dielectric permittivity for F=0 N and B≠0 T. and $\varepsilon^{\prime }\left(0\right)$ is the relative dielectric permittivity for F=0 N and B=0 T.

The relative contribution $A_{\varepsilon^{\prime }}$ brought by the force *F* to the apparent increase of the relative dielectric permittivity of the composites in a magnetic field, superimposed on the field of mechanical deformations is defined by:(6)$$A_{\varepsilon^{\prime }}=\left[\frac{\varepsilon^{\prime }{\left(B\right)}_{F}}{\varepsilon^{\prime }{\left(B\right)}_{0}}-1\right]\times 100$$
where $\varepsilon^{\prime }{\left(B\right)}_{F}$ and $\varepsilon^{\prime }{\left(B\right)}_{0}$ are the relative dielectric permittivities of the composites in a magnetic field with, and respectively without the influence of the field of mechanical deformations. By using the data $\varepsilon^{\prime }=\varepsilon^{\prime }{\left(B\right)}_{F=0N}$ from Figure 9 one obtains the *MDE* as shown in Figure 10a. Similarly, the relative contribution of $A_{\varepsilon^{\prime }}$ are shown in Figure 10b,c. The obtained results show that the values of $\varepsilon^{\prime }$ can be controlled by the quantity of Fe_3_O_4_ particles, by the applied forces *F* or by the magnetic flux density *B*.

The electrical conductivity *σ* of the composites in the electric field of frequency *f* is calculated by $\sigma =2\pi f\varepsilon_{0}\varepsilon^{\prime }D_{S}$. Thus, by using Equation (4) and $\varepsilon_{0}=8.854\;\mathrm{pF}/m$ one obtains $\sigma =8\pi 10^{-9}f\left(\mathrm{kHz}\right)C_{S}\left(\mathrm{pF}\right)D_{S}/9$. By using the variations $C_{S}=C_{S}{\left(B\right)}_{F}$ and $D_{S}=D_{S}{\left(B\right)}_{F}$ from Figure 6 and Figure 7, respectively, one obtains the variation $\sigma =\sigma {\left(B\right)}_{F}$, as shown in Figure 11.

The results show that the electrical conductivity increases with the increasing of magnetizable phase volume fraction $\Phi_{{\mathrm{Fe}}_{3}O_{4}}$, and for a fixed $\Phi_{{\mathrm{Fe}}_{3}O_{4}}$, the electrical conductivity increases with increasing compression forces *F*.

In order to quantify the influence of the magnetic field at *F* = 0 N on the values of *σ*, for each quantity of ${\mathrm{Fe}}_{3}O_{4}$ from the ML, we introduce the concept of relative and apparent magneto-conductive effect as:(7)$$MCE\left(\%\right)={\left[\frac{\sigma \left(B\right)}{\sigma \left(0\right)}-1\right]}_{F=0N}\times 100$$
where $\sigma \left(B\right)$ and $\sigma \left(0\right)$ are the electrical conductivities of MC composites in the presence, and respectively in the absence of the magnetic field.

The relative contribution $A_{\sigma }$, brought by the values of *F* to the increase of electrical conductivity *σ* of the composites in the magnetic field, is quantified by:(8)$$A_{\sigma }=\left[\frac{\sigma {\left(B\right)}_{F}}{\sigma \left(B\right)}-1\right]\times 100$$
where $\sigma {\left(B\right)}_{F}$ and $\sigma \left(B\right)$ are the electrical conductivities of MC composites in the presence of a magnetic field with, and respectively without the presence of the mechanical deformations field.

By using the variation $\sigma {\left(B\right)}_{F=0N}$ from Figure 11 one obtains the variation $MCE=MCE{\left(B\right)}_{F=0N}$ as shown in Figure 12a. Then, by introducing in Equation (8) the functions $\sigma =\sigma {\left(B\right)}_{F\neq 0N}$, and $\sigma =\sigma {\left(B\right)}_{F=0N}$ from Figure 11, we obtain the variation $A_{\sigma }=A_{\sigma }{\left(B\right)}_{F\neq 0N}$, as shown in Figure 12b,c.

The quantity *MDE* defined in Equation (5) provides a measure of the relative apparent increase of the equivalent electrical capacitance of ED in a magnetic field. However, the quantity *MCE* defined in Equation (7) is a measure of the relative apparent decrease of the equivalent electrical resistance of ED in a magnetic field. It is well known that the ED fabricated by using magnetizable composites in a magnetic field can be represented, from an electrical point of view, through a network of electrical microcapacitors *C*_z_ together with a network of electrical microresistors *R*_z_, electrically interconnected in series or parallel [44,51]. The values of *C*_z_ and *R*_z_ are influenced by the distance *z* between the center of masses of the magnetizable particles [41,51], in our case the mass centers of magnetite nanoparticles. The quantity *z* is given by [51]:(9)$$z=\delta [1-\left(3\pi d_{{\mathrm{Fe}}_{3}O_{4}}^{2}B^{2}/4\mu_{0}\delta k\right)]$$
where $\delta \equiv d_{{\mathrm{Fe}}_{3}O_{4}}\Phi_{{\mathrm{Fe}}_{3}O_{4}}^{-1/3}$ is the initial distance between the center of masses of magnetic nanoparticles, $d_{{\mathrm{Fe}}_{3}O_{4}}$ is the average diameter of magnetite nanoparticles, k is the coupling constant between the magnetizable phase and cotton microfibers.

One can note that by decreasing the values of z with increasing $\Phi_{{\mathrm{Fe}}_{3}O_{4}}$, at fixed values of *B*, or by increasing the values of *B*, the quantities *C*_z_ increase, while *R*_z_ decrease. These observations have important effects on both *MDE* and *MCE*.

Thus, for the MC_1_ composite, *MCE* increases linearly with *B* up to about 315 mT, after which the increase is quasilinear, due to the resistance of microfibers from within the fabric. However, the quantities *MDE* increase linearly with *B* up to about 315 mT, where a maximum value is attained. However, for *B* ≳ 315 mT, *MDE* decreases with increasing *B* due to the agglomeration of the nanoparticles and due to the resistance induced by the microfibers in the cotton fabrics. As a result, the mass increases, and in order to bring the nanoparticles together, a higher value of the magnetic interactions is required. However, this process cannot be achieved in the range of the used values of *B*. 

The obtained effect is presented in Figure 10a and shows that *MDE* decreases with the increase of *B* in the range from 315 to 390 mT.

The functions *MCE* for the composites show different behavior. Thus, for the MC_1_ composite the contact electrical resistance is high when 0 ≤ *B* (mT) ≤ 98. The effect is due to the fact that in this range, the magnetic interaction intensity between the dipoles is small as compared to the resistance force opposed by the microfibers to the movement of magnetite nanoparticles. For *B* ≥ 100 mT, the intensity of magnetic interactions between dipoles is higher than the value of the resistance force generated by the microfibers. The net effect is an increase of *MCE* when increasing *B*. 

In the case of the MC_2_ composite, due to the doubling of the magnetic force, the intensity of magnetic interactions is higher than the resistance one, which is opposed to the movements of the magnetic dipoles. At *B* = 315 mT, the *MCE* has a maximum, after which the values of *MDE* decrease with increasing *B*. By increasing $\Phi_{{\mathrm{Fe}}_{3}O_{4}}$, the magnetite nanoparticles agglomerate further, their mass increases and the movement of the agglomerates through the microfibers is slowed down. For 315 ≤ *B*(mT) ≤ 390, the increased mass of the agglomerates together with the resistance force leads to an increase of the contact resistance. The net effect is a decrease of *MCE* with increasing *B*, in this range. 

The agglomeration of magnetite nanoparticles can appear as an effect of spontaneous magnetization [52] and also by the agglomeration of nanoparticles due to the action of the forces of electrostatic interaction of particles [53]. The movement of the formed assemblies through the microfibers of the fabric is slowed down.

Thus, at values of *B*, from 315 mT to 390 mT, as an effect of the increase in the mass of nanoparticles and as a result of the increase in the strength of resistance of textile microfibers, the contact resistance between nanoparticles and microfibers increases. The effect obtained (see Figure 12a) is the decrease of *MCE* with the increase of *B* for the range between 315 mT and 390 mT.

It is known that in a gravitational field, and respectively by application of a uniform field of compression forces, the distance between magnetizable particles decreases [54,55]. The net effect is the increase of the equivalent electrical capacitance, and respectively a decrease of the equivalent electrical resistance of ED with increasing the compression forces. Similar effects are reported in Refs. [54,55] and it can be observed from the above equations for the equivalent electrical capacitance and resistance. These also show that fixed values of *B* and $\Phi_{{\mathrm{Fe}}_{3}O_{4}}$, the values of $A_{\varepsilon^{\prime }}$ and $A_{\sigma }$ increase linearly with *F*, as seen from Figure 10b and Figure 10c, and Figure 12b and Figure 12c, respectively.

The induced electrical effects by the magnetic field in the obtained composites are due to the magneto-deformations, as described in refs. [34,35,36,55]. In these references, the electrical response of the devices manufactured using composites based on CI microparticles is sensibly higher, as compared to the electrical devices obtained here. The reason for this difference is the usage of CI microparticles at high values of volume fractions of the magnetizable phase. However, the stable electrical response with time and the tunable values of the ED obtained here, make them better candidates for the fabrication of magneto-tactile electrical devices based on cotton fibers and magnetic liquid.

## 4. Conclusions

Magnetic composites were fabricated, based on cotton fibers soaked with magnetic liquid. The obtained composites are then used as dielectric materials for the fabrication of mechanically stable electrical devices (magneto-tactile sensors). By using an original experimental setup, we measured the electrical capacitance (*C*_S_) and the loss tangent (*D*_S_) in a medium-frequency electric field, *f* = 1 kHz, superimposed on a static magnetic field with densities 0 ≤ *B* (mT) ≤ 400, and under uniform mechanical compression forces of 0, 4 and 8 N. From the obtained data we calculated the relative dielectric permittivity $\varepsilon^{\prime }$ and electrical conductivity *σ* of the composites as functions of *B* and *F*. The results show that the electrical conductivity increases with the quantity of magnetizable phase, while for a fixed amount of the magnetic fluid, the electrical conductivity increases with compression forces *F*. It is shown that both $\varepsilon^{\prime }$ and σ can be fixed through the values of force *F*, and finely tuned through the density *B*. These properties make the new composite a promising candidate for the fabrication of low-cost magneto-tactile sensors, much more stable compared to those reported in our previous work for devices manufactured using composites based on CI microparticles.

## Figures and Tables

**Figure 1 materials-16-03222-f001:**
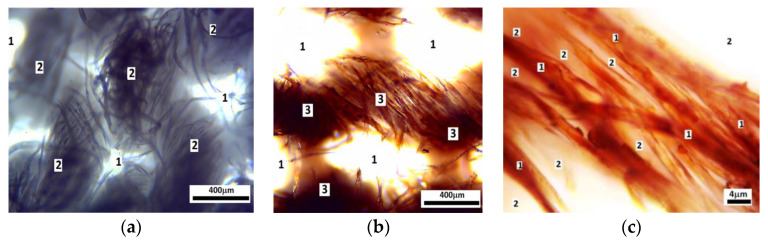
Optical microscopy of: (**a**) Gauze bandage GB. 1—empty space between cotton threads, 2—microfibers. (**b**) Whole composite material. 1—empty space between cotton threads, 3—microfibers impregnated with magnetic liquid ML. (**c**) Fabric’s thread. 1—microfibers impregnated with ML (at a higher magnification), 2—interstitial space.

**Figure 2 materials-16-03222-f002:**
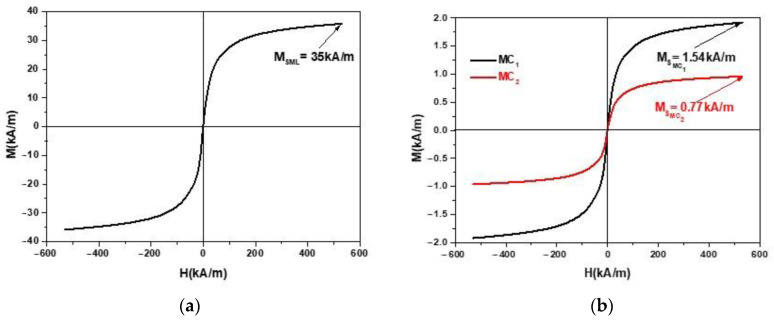
Magnetization *M* as a function of intensity *H* of the magnetic field, for: (**a**) ML. (**b**) MC.

**Figure 3 materials-16-03222-f003:**
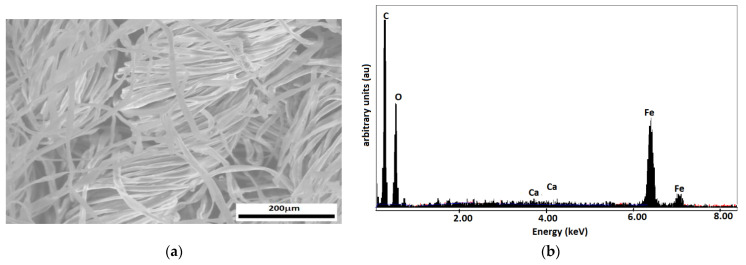
(**a**) SEM image of cotton fibers with ML. (**b**) Spectral analysis of cotton fabric with ML.

**Figure 4 materials-16-03222-f004:**
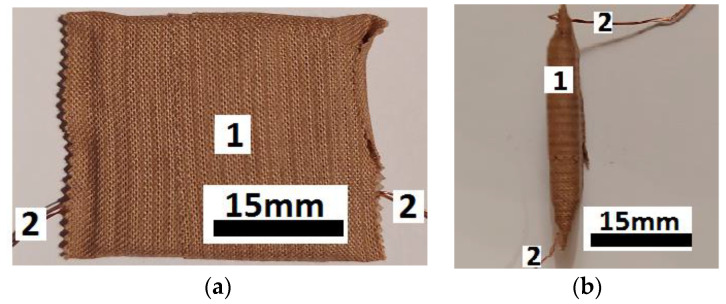
(**a**) Electrical device ED: Top-view. (**b**) Electrical device ED: Cross-sectional view. 1—PTS tape, 2—copper conductors.

**Figure 5 materials-16-03222-f005:**
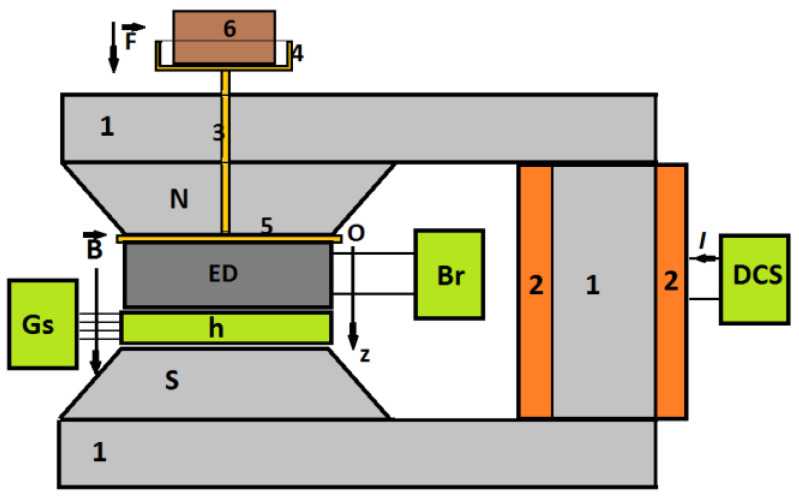
Overall setup. 1—magnetic core, 2—coil, 3—non-magnetic spindle, 4—non-magnetic plate, 5—the non-magnetic disk, 6—non-magnetic marked masses, N and S—magnetic poles, ED—electric device, Br—RLC bridge, Gs—gaussmeter, h—Hall probe, DCS—continuous source current, Oz—coordinate axis, **B**—magnetic flux density vector, **F**—force vector, *I*—the intensity of the electrical current.

**Figure 6 materials-16-03222-f006:**
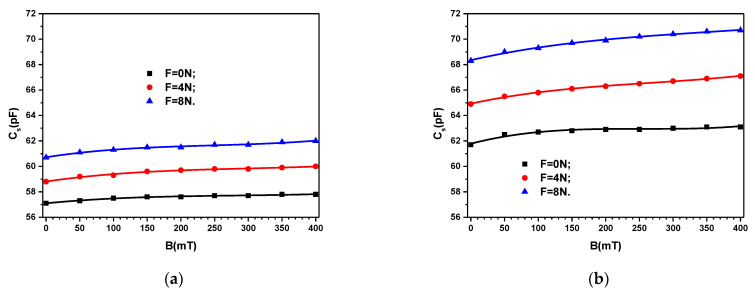
Electrical capacitance Cs of the devices ED as a function of values *B* of the magnetic flux density superimposed on the electric field with the frequency *f* = 1 kHz, and values *F* of the force as a parameter. (**a**) MC_1_. (**b**) MC_2_.

**Figure 7 materials-16-03222-f007:**
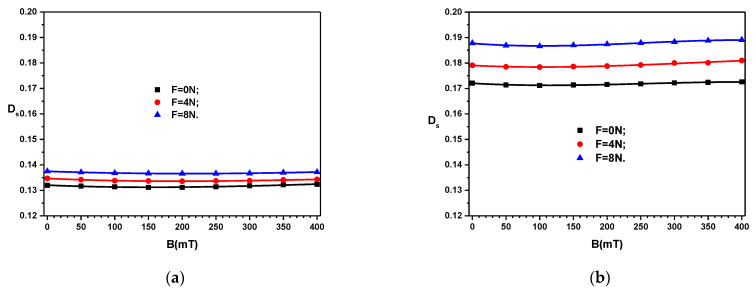
Loss tangent *D*_S_ of the devices ED as a function of values *B* of the magnetic flux density superimposed on the electric field with the frequency *f* = 1 kHz, and values *F* of the force as a parameter. (**a**) MC_1_. (**b**) MC_2_.

**Figure 8 materials-16-03222-f008:**
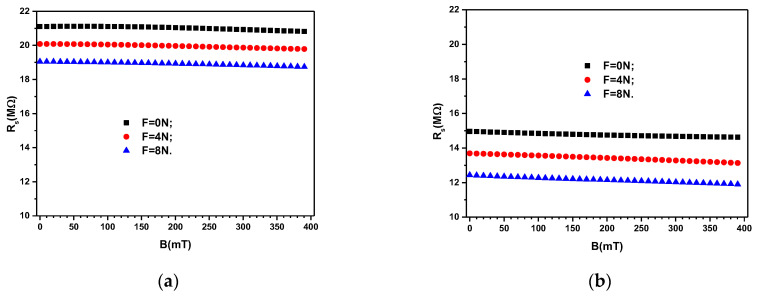
Electrical resistance *R*_S_ of the devices ED as a function of values *B* of the magnetic flux density superimposed on the electric field with the frequency *f* = 1 kHz, and values *F* of the force as a parameter. (**a**) MC_1_. (**b**) MC_2_.

**Figure 9 materials-16-03222-f009:**
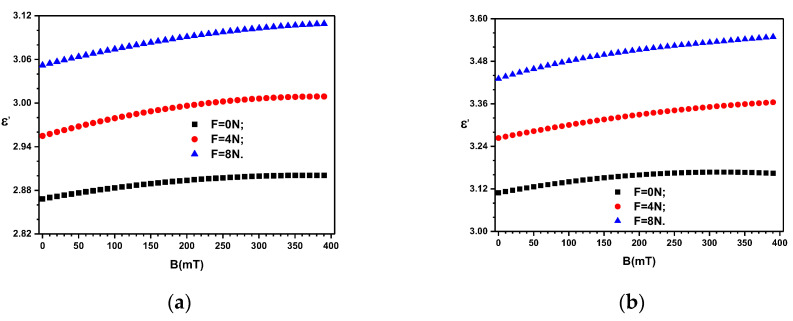
Relative dielectric permittivity *ɛ*′ of the devices ED as a function of values *B* of the magnetic flux density superimposed on the electric field with the frequency *f* = 1 kHz, and values *F* of the force as a parameter. (**a**) MC_1_. (**b**) MC_2_.

**Figure 10 materials-16-03222-f010:**
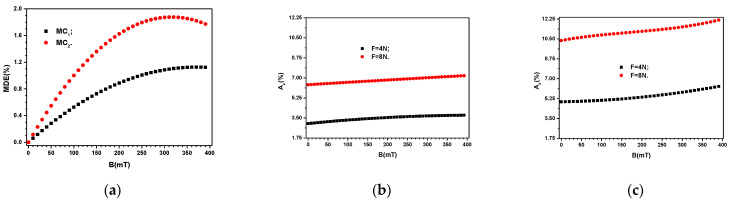
(**a**) Relative apparent *MDE* for MC_1_ and MC_2_ when *F* ≠ 0*N*F. (**b**) The relative contribution *A*_ɛ^′^_ for MC_1_. (**c**) The relative contribution *A*_ɛ^′^_ for MC_2_.

**Figure 11 materials-16-03222-f011:**
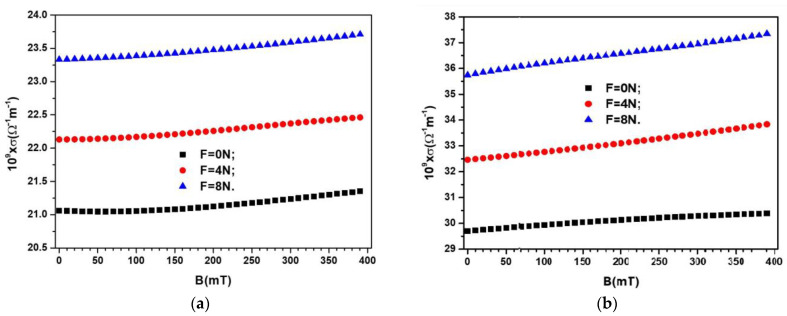
Electrical conductivity σ of the devices ED as a function of values *B* of the magnetic flux density superimposed on the electric field with the frequency *f* = 1 kHz, and values *F* of the force as a parameter. (**a**) MC_1_. (**b**) MC_2_.

**Figure 12 materials-16-03222-f012:**
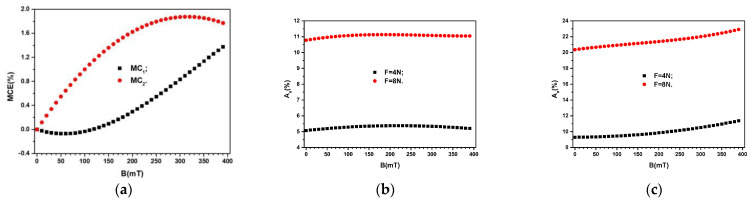
(**a**) Relative apparent *MCE* for MC_1_ and MC_2_ when F≠0NF. (**b**) The relative contribution $A_{\sigma }$ for MC_1_. (**c**) The relative contribution $A_{\sigma }$ for MC_2_.

**Table 1 materials-16-03222-t001:** Masses (*m*) and volumes (*V*) of the components of the two composite materials.

Sample	*m*_GB_ (g)	*m*_ML_ (g)	*V*_GB_ (cm^3^)	*V*_ML_ (cm^3^)
MC_1_	0.360	0.138	0.348	0.114
MC_2_	0.360	0.276	0.348	0.228

**Table 2 materials-16-03222-t002:** Volumes (*V*) and volume fractions (*Φ*) of the components of the two composite materials.

Sample	*V*_oil_ (cm^3^)	*V*_f_ (cm^3^)	*V*_Fe_3_O_4__ (cm^3^)	*Φ*_oil_ (vol.%)	*Φ*_f_ (vol.%)	*Φ*_Fe_3_O_4__ (vol.%)
MC_1_	0.10659	0.234	0.00741	30.6	67.2	2.2
MC_2_	0.21318	0.120	0.01482	61.6	34.5	4.4

## Data Availability

The data are available from Authors, by reasonable requests.

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
