# Peer review of "Composites Based on Cotton Microfibers Impregnated with Magnetic Liquid for Magneto-Tactile Sensors"

_materials, 2023, doi:10.3390/ma16083222_

Round 1

Reviewer 1 Report

The article is devoted to research of properties and application potential of composite magnetoactive materials. This is a very promising direction in the development of engineering and technology, and therefore the results obtained in the course of such studies are given close attention.

The studies presented in the paper are of scientific interest, but there are a number of comments and recommendations:

1. The percentage of authors self-citing reaches 27%. It is probably worth expanding the range of analyzed studies, including those in related materials, which will make it possible to study the material more qualitatively.

2. The abstract in the work needs significant adjustment.

3. Figure 2 needs replacement/correction. The image is fuzzy, detail is low.

4. The authors assume spontaneous magnetization. She is not spontaneous. Agglomeration can occur due to the action of the forces of electrostatic interaction of particles, leading to the appearance of secondary magnetization and self-organization of particles. You can read about the formation of agglomerates here (https://doi.org/10.3390/ma14092376), as well as in other works.

5. Conclusions are weakly connected with the main part and require processing.

The authors should reveal in more detail the concept of the final device, the development of which is directed by research. One gets the impression that they have little idea of him, and we are talking about a nebulous vague prospect.

Reviewer 2 Report

In this work, the authors present a comprehensive investigation into the dielectric properties of cotton fiber-based magnetic composites, which could potentially provide a low-cost route for magneto-tactile sensors. However, there are significant issues present in the current manuscript that require major revisions before publication:

·        Regarding the setup, it is evident that the electronic device (ED) is not airtight. Could the authors comment on how they excluded the impact of the air gap in their experiments? For example, capacitance increase with increasing force in figure 5 could also be coming from decreasing air gap distance between electrodes, in addition to the interaction between magnetic particles. Additionally, the side view image in figure 3(b) shows the thickness of ED is about 5 mm, much thicker than the sum of copper plates plus composite (0.8-2 +0.4 = 2 mm). Although the thickness of the PTS tape is not specified, the thick side view could indicate the existence of an air gap in the ED.

·        The magnetic response data in figure 5 also requires more careful examination: have the authors checked how the capacitance values evolve as a function as s function of idle time with no field applied? Also, suppose the capacitance change is genuinely from magnetic interaction, then in that case it should be symmetrical – have the authors reserve the direction of the field and check whether the toggle is still the same?

·        How is the volume of magnetic liquid calculated in table 1? Is it merely calculated using mass divided by its density in line 47? If so, could the authors justify why ML has the same density when incorporated into the cotton fiber?

·        In the experimental setup section, it was stated that the magnetic field is tunable is steps of 50 mT. Why does data in figure 7 and 8 show steps around 10 mT, much denser than that in figure 5 and 6?

·        In line 82 the saturation magnetization of the composite should be MS_MC instead of MS_ML.

Reviewer 3 Report

The authors produced an original composite material based on cotton fabric impregnated with an oil-based magnetic fluid. dielectric permittivity and electrical dependencies

The conductivity of the magnetic field strength and sample compression was investigated in the paper. The authors made an original experimental setup to study these dependencies.

The work contains original material, a new magnetically active composite is being studied on the original setup. However, the theoretical interpretation of the obtained results and comparative analysis need to be improved.

Several comments can be made on the article:

1. The authors did not justify the choice of an oil-based magnetic fluid. Why not water or kerosene, which absorb better and evaporate faster.

2. Why the authors rolled the fabric into a roll, and did not fold it in folds, this method gave a greater change in the dielectric parameters.

3. The authors use two samples, but none of the obtained graphs and results compare the obtained data in terms of the structure and properties of the samples.

4. The authors did not indicate the measurement error. What is the repeatability of the results? Is there a creasing of the sample and a change in its parameters.

5. Not all designations in the formulas have a decoding.

6. In conclusion, it is necessary to compare the obtained data with the results obtained using other sensors.

After these minor changes, the article can be published.

Reviewer 4 Report

The current work focuses on the composites based on cotton microfibers and magnetic liquid: Effects of the magnetic field superimposed on the medium-frequency electric field and uniform mechanical tension on the relative dielectric permittivity and electrical conductivity. The author’s great effort into the manuscript but some issues should be addressed. 

Title 

The title is very long, rephrase it with an informative target

Abstract

-Magnetic fiber has been extensively studied. First, show the novelty of the current work and then the main outputs.

Introduction

- The introduction is very short and doesn’t provide sufficient background and the most relevant references are not included. 

-The novelty of this work is not highlighted and the author's contribution was unclear compared to other previous works. 

Results and Discussion 

-One of the main problems in the manuscript is that the authors only show results without interpretations or confirmation by citation. More details are required to explain the obtained results.

-The figure for magnetic properties of the composites e.g. VSM should be inserted to clear the behavior to the reader.

Round 2

Reviewer 1 Report

The main comments formulated by the reviewers were taken into account by the authors in the revised material. The article may be published in its present form.

Reviewer 2 Report

The authors provided great feedback on my questions and addressed all my concerns. I don't have any further comments.